# Investigation of Functional Connectivity Differences between Voluntary Respirations via Mouth and Nose Using Resting State fMRI

**DOI:** 10.3390/brainsci10100704

**Published:** 2020-10-03

**Authors:** Ju-Yeon Jung, Chan-A Park, Yeong-Bae Lee, Chang-Ki Kang

**Affiliations:** 1Department of Health Science, Gachon University Graduate School, Incheon 21936, Korea; 9955me@gc.gachon.ac.kr; 2Biomedical Engineering Research Center, Gachon University, Incheon 21936, Korea; chaoskit21c@gmail.com; 3Department of Neurology, Gil Medical Center, Gachon University College of Medicine, Incheon 21565, Korea; yeongbaelee@gmail.com; 4Neuroscience Research Institute, Gachon University, Incheon 21565, Korea; 5Department of Radiological Science, College of Health Science, Gachon University, Incheon 21936, Korea; 6Department of Health Sciences and Technology, Gachon Advanced Institute for Health Sciences & Technology, Gachon University, Incheon 21936, Korea

**Keywords:** voluntary respiration, mouth breathing, resting state fMRI, functional connectivity

## Abstract

The problems of mouth breathing have been well-studied, but the neural correlates of functional connectivity (FC) still remain unclear. We examined the difference in FC between the two types of breathing. For our study, 21 healthy subjects performed voluntary mouth and nasal breathing conditions during a resting state functional magnetic resonance imaging (fMRI). The region of interest (ROI) analysis of FC in fMRI was conducted using a MATLAB-based imaging software. The resulting analysis showed that mouth breathing had widespread connections and more left lateralization. Left inferior temporal gyrus had the most left lateralized connections in mouth breathing condition. Furthermore, the central opercular cortex FC showed a significant relationship with mouth breathing. For nasal breathing, the sensorimotor area had symmetry FC pattern. These findings suggest that various FCs difference appeared between two breathing conditions. The impacts of these differences need to be more investigated to find out potential link with cognitive decline in mouth breathing syndrome.

## 1. Introduction

Respiration is the process of taking in oxygen required for cellular metabolism from the outside environment and releasing carbon dioxide from the cells to the external environment. The act of breathing by human lungs, comprised of inhalation and exhalation, is an active movement that facilitates physiological respiration. Breathing in and out through the nose is referred to as nasal breathing, which plays various vital roles, owing to the hair present in the nostrils and mucous in the nasal cavity. These roles aid in filtering unwanted particles and preventing them from entering the lungs, and control air temperature and humidity inside the trachea. In spite of the advantages of nasal breathing [1], humans often resort to breathing through their mouths, especially in cases of nasal airway obstruction [2,3].

Mouth breathing is a multifactorial problem that can be attributed to physiological and mechanical etiologies [4]. It affects 17% of the general population [5] and 53.3% of children [6]. The lack of filtering, humidifying, and warming of air inhaled through the mouth may lead to decreased lung function [7]. In addition, inefficient O_2_ and CO_2_ exchange is produced during mouth breathing, since there may be different concentration level of nitric oxide (NO). NO is more produced in the sinuses located in the nasal pathway [8]. As a potent vasodilator, NO reaches the lungs and it is diffused into the capillaries of the alveoli, expanding vessels, and increasing O_2_ and CO_2_ exchange [8,9]. Therefore, the insufficient NO concentration of mouth breathing had some problems when it came to absorbing O_2_ during gas exchange. A large amount of air incoming from mouth breathing can also disturb transferring of O_2_ from the lung to hemoglobin [9]. Consequently, mouth breathing can negatively affect not only the musculoskeletal system, but also cognitive function; the effect of mouth breathing on the brain has already been reported by several studies [10,11,12,13,14,15].

A previous study reported that the degree of oxygen exchanges in the prefrontal cortex is higher during mouth breathing than during nasal breathing [16]. The study also reported an increased oxygen load during mouth breathing as compared to nasal breathing. Furthermore, the associated signal changes between nasal and mouth breathing conditions appear to result in different activation patterns in the brain [17]. Specifically, increased pre- and post-central activation has been observed in mouth breathing when compared to nasal breathing. These findings may support a role for different brain functional connectivity (FC) for different breathing methods. Surprisingly, no information is currently available on FC changes for the two different voluntary breathing methods.

Brain system fluctuated slowly and synchronously without any explicit task. The function of the resting state network (RSN) remains a controversial issue, but generally self-referential thinking, emotional processing and recalling memories were known as related functions of RSN [18]. The human brain consumes 60–80% of the total energy in the resting state, but the increase in energy consumption in the active or task related state is less than 5% [19]. Although task-related functional magnetic resonance imaging (fMRI) results during mouth breathing have been reported [20], there are few studies that analyze FC changes in the resting state human brain.

Although many people are forced to perform mouth breathing due to various physical and physiological conditions, the effects of mouth breathing on brain function have not been well-documented. A suitable resting of the brain is an important process for recovery and activation of the brain, and breathing is an important factor that can influence the resting state. The brain activity changed in the resting state can also affect the energy metabolism of the brain. Therefore, it is important to define mouth breathing impacts on resting state FC for performing proper rest and various brain functions. Here, we explored whether there exist different effects of mouth and nasal breathing on neural correlates of FC in a voluntary resting state condition.

## 2. Materials and Methods

### 2.1. Subjects and Data Acquisition

Twenty-two healthy subjects (ten men and twelve women; mean age, 22.27 ± 1.42 years) participated in the study and provided written informed consent, but one subject was excluded for the analysis due to the large global signals (GS) variance. The study protocol was approved by the institutional review board (IRB number: GDIRB2017-174). The procedure was performed in accordance with approved guidelines. Subjects had no history of neurological, psychiatric, or respiratory disorders.

The experiment was performed using a 3T MRI scanner (Siemens Verio, Erlangen, Germany), with a commercially available 12-channel radio-frequency (RF) head matrix coil for whole-brain imaging. All participants underwent three imaging sequences (one structural and two resting-state scans) in scanner. The first was a high-spatial-resolution T1-weighted anatomical imaging sequence of three-dimensional (3D) magnetization–prepared rapid acquisition gradient echo (MP-RAGE). The second was a blood-oxygen-level-dependent (BOLD) fMRI sequence of two-dimensional (2D) echo planar imaging (EPI) with repetition time (TR)/echo time (TE) = 2000 ms/30 ms, field of view (FOV) = 192~195 mm, 3 × 3 mm^2^ (or 3.02 × 3.02 mm^2^) in-plane resolution, 3 mm slice thickness (50% distance factor, 1.5 mm gap) and 30 interleaved imaging slices, in order to cover the whole brain (135 mm in z-axis), including the cerebellum and flip angle (FA) = 90°.

Every participant underwent a mouth breathing familiarization protocol for about 20 min before beginning the session. This protocol was performed to eliminate the unnatural process of mouth breathing control (like deep breathing, expanded lung volume, breathing frequency), acquiring the normalized respiration-related physiological data by GS [21,22]. Furthermore, the subjects were made to use nose plugs to induce mouth breathing naturally, without any interference or unnecessary cognition.

The participants were asked to breathe voluntarily only through their mouth or nose at the beginning of each session. They were instructed to keep their heads without movement, keep their eyes closed and stay awake throughout the scan. They wore the nose plugs in their nostrils during the mouth breathing condition in order to prevent nasal breathing, and removed the nose plugs during the nasal breathing condition to induce natural breathing with their mouth closed. The order of the breathing condition was randomized across subjects. For each session, we collected 154 dynamic data volumes for about 5 min.

### 2.2. Functional Connectivity Processing

CONN is a MATLAB-based platform software for the calculation, display, and analysis of resting state FC in fMRI. It was used to calculate the strength and significance of the bivariate correlation among region of interest (ROI) pairs within all subjects’ data in the mouth and nasal breathing conditions. 

Preprocessing for the FC analysis was performed using the CONN (version 18b) FC toolbox [23]. Preprocessing included the removal of the first four volumes, realignment to the first volume for head-motion correction, co-registration, segmentation, normalization to the EPI template with a resampling voxel size of 2 × 2 × 2 mm^3^, and smoothing with an 8 mm full-width half-maximum Gaussian kernel. We used an established strategy for spatial and temporal preprocessing to define and eliminate confounds in the BOLD signal, to prevent the effects of movement and physiological noise factors in the data [24]. CONN uses the component-based noise correction method (CompCor strategy) for the noise reduction in both BOLD and perfusion-based fMRI. During spatial and temporal preprocessing, the confounds are eliminated in BOLD signal to prevent the effects of movement and physiological noise factors in the data [24]. After the CompCor processing, temporal standard deviation is reduced, as compared to that before the CompCor. The CompCor processing step could help to increase the activation of voxel. We then used the CONN toolbox to implement a denoising process on confounding factors using linear regression and band-pass filtering (0.008–0.09 Hz) to remove the subject’s estimated motion parameters (6 motion parameters and 6 first-order temporal derivatives), including BOLD signals in white matter and cerebrospinal fluid regions, which were included as additional covariates.

CONN used a seed-based resting state FC analysis. The analysis of FC followed 2 steps. First, seed-based analysis was performed in first level analysis for each subject to obtain the result of connectivity between individual seed to voxel or ROI. Pearson’s correlation coefficient was calculated between the 164 seeds time course and all other voxels. Second, the second level analysis between groups was performed to analyze the contrast between conditions. After first level analysis, ROI level analyses were performed through F—or Wilks lambda statistics. To perform weighted general linear model analysis at the second level, Fisher’s transformation was used to convert correlation coefficients [25].

In the CONN toolbox, a total of 164 ROIs were used as the seeds. Among them, 132 ROIs were atlas of cortical and subcortical areas from the fMRI of the brain software library (FSL) Harvard-Oxford atlas, as well as cerebellar areas from the automated anatomical labeling (AAL) atlas; 32 ROIs were atlases of networks (i.e., 32 ROIs across 8 networks). The networks included default mode network (DMN), sensorimotor network, visual networks, salience networks, dorsal attention networks, frontoparietal networks, language networks, cerebellar networks. RSN and ROIs were each independently analyzed, and FC pairs were obtained with 164 ROIs, in which all of the RSN ROIs and atlas ROIs were combined (Appendix A).

### 2.3. Laterality of Functional Connectivity

We evaluated which side of hemisphere had higher pairwise connection strength. The laterality of functional connections was assessed as differences between left and right hemisphere. T-score (ROI connectivity strength values) was used to statistically compare the values in the contralateral hemisphere using the laterality index (LI) formula (LI = (T_left_ − T_right_)/(T_left_ + T_right_)) [26]. According to LI, the positive values higher than 0.2 indicate left lateralization and the negative values lower than -0.2 indicate right lateralization (leftward: values > 0.2; rightward: values < −0.2).

### 2.4. Statistical Analysis

Seed-based FC measures including ROI-to-ROI and seed-to-voxel analyses were conducted to create ROI-to-ROI connectivity. We used a bivariate correlation and hemodynamic response function (HRF) weighting to calculate FC. Connectivity contrast thresholds between all ROI sources were calculated alongside uncorrected *p*-values and FDR-corrected *p*-values for second level analysis. F-test was used to calculate the multivariate connectivity strength for each seed.

In ROI-to-ROI analysis, we tested the Mouth > Nose and Mouth < Nose contrasts to create connections between whole ROIs and conducted networks analysis with one-sided inferences. The connectome ring maps were expressed to include threshold ROI-to-ROI connection by intensity and threshold seed ROIs (F-test) with one-sided positive seed level correction and permutation tests. The significance of ROI-to-ROI connection by intensity was determined through the FDR-corrected *P* < 0.05 with seed-level correction. The significance of seed ROIs (F-test) was determined through uncorrected *P* < 0.05.

## 3. Results

In comparison of GS variance, there were no significant differences between mouth and nasal breathing signals (*P* > 0.05) (Appendix A). Two types of FC images were displayed. FC as connectome ring type with ROI in reference brain option was shown in Figure 1. Moreover, 3D rendering maps were displayed in Figure 2 and Figure 3, where four views of (a) dorsal, (b) anterior, (c) right, and (d) left were shown for easier understanding of spatial location. The ROI-to-ROI FC connectome ring (Figure 1) showed more prominent connections in the mouth breathing condition than in the nasal breathing one. As shown in the 3D rendering maps in Figure 2 and Figure 3, more densely connected ROIs were present in the mouth breathing condition. As shown in Table 1, Mouth > Nose contrast had 22 significant ROI-to-ROI connections, of which 6 seeds that had significantly strong connectivity (Table 2). The left and right lateral sensorimotor area had the most significant connection in the Mouth > Nose contrast (*P* < 0.05, *T* = 6.02) (Table 1). The left sensorimotor lateral area had significant connection with 4 ROIs (*P* < 0.05, intensity = 18.11) and right sensorimotor lateral area had significant connection with 5 ROIs (*P* < 0.05, intensity = 20.21). The right central opercular cortex had the most significant differences among connected seeds (*P* = 0.0015, *F* = 6.73) in Mouth > Nose contrast. The intensity and size results are alternative ways to assess whether the connectivity between the seeds and all other ROIs show any significant effects. The size measurement represents the number of above-threshold connections between the seed and all other ROIs, and the intensity represents the overall strength as the sum of the absolute T-values for these above-threshold connections [25].

In the Mouth < Nose contrast, we observed 16 significant stronger ROI-to-ROI connections and had stronger connectivity in 7 seeds (Table 3 and Table 4). The left sensorimotor lateral and sensorimotor superior had the most significant connection for this contrast (*P* < 0.05, *T* = 6.85) (Table 3). The sensorimotor superior had the biggest intensity value (*P* < 0.05, intensity = 34.21) and seven connected ROIs (*P* < 0.05, size = 7) (Table 4), which was the highest number of connection pairs between other ROIs, including the left and right lateral sensorimotor area and left postcentral gyrus, left and right precentral gyrus, and left and right central opercular cortex (Table 3).

We calculated the connectivity *T*-statistics on Mouth > Nose contrast by using laterality index. We discovered that four pairs were significantly left lateralized, whereas the other pairs seemed symmetric (Table 5). Especially, four left lateralized pairs were the connections with inferior temporal gyrus (ITG).

## 4. Discussion

In this study, we investigated whether mouth breathing has different effects on FC as compared to nasal breathing. The results showed that more connectivity patterns appeared during the mouth breathing condition as compared to nasal breathing. Nasal breathing showed symmetrically activated on left and right hemisphere; mouth breathing showed left lateralized connections. Overall, we assessed different connectivity patterns between mouth and nasal breathing conditions and analyzed the significantly important seeds, including the sensorimotor area, central opercular cortex and lateral parietal cortex in the DMN.

### 4.1. Asymmetric Functional Connectivity Pattern

Although the Mouth > Nose contrast showed left lateralized asymmetry patterns, the Mouth < Nose contrast showed more symmetry patterns (Figure 2 and Figure 3). The Mouth < Nose contrast had 6 of 16 symmetrically connected pairs between the left and right hemispheres (Table 3). Symmetrical connectivity means that one seed (sensorimotor superior) and its connected ROIs were observed symmetrically in both hemispheres. The sensorimotor superior was the seed most connected to other ROIs at Mouth < Nose; it was observed that 6 ROIs of both hemispheres were correlated at the same time. In the resting state networks of normal subjects, it was observed that functional links were symmetrically formed in both hemispheres [27], and it is considered that similar patterns were observed in Mouth < Nose.

In Mouth > Nose contrast, the seeds of the left hemisphere had 9 connected pairs (Table 2). Furthermore, only left laterality pairs were discovered on both hemispheres (Table 5). We suggested that these connections were related with language process. When speaking or vocalizing, mouth breathing is accompanied according to the phrase, and exhalation is performed through the oral cavity [28]. To maintain breathing, inhalation is performed through the oral cavity or nasal cavity. Furthermore, the speech processing has a lot of influence in the left hemisphere, as it is involved in Broca and Wernicke’s areas. It is thought that stimulation by mouth breathing induces connectivity of the left hemisphere like speech. According to our results, left inferior temporal gyrus connections which have significantly left lateralized connections were appeared in Mouth > Nose contrast (Table 5). Left inferior temporal gyrus known as important structure for language process had linked with left central opercular cortex and left insular cortex, which also have a role in language processing [29,30,31]. In addition, the connections with cerebellum have been described as related areas of respiratory information processing [32]. It is possible that differences in mouth sensation and oral muscle contraction during mouth breathing could contribute to change function in left hemisphere regions. Thus, these findings may indicate dominant functions of the left hemisphere, which is concerned with language process and respiratory sensation of breathing.

### 4.2. Sensorimotor Area

In the present study, most powerful intensity of the ROI pairs appeared at sensorimotor areas in both Mouth > Nose and Mouth < Nose contrast (Table 2 and Table 4). These findings suggest that both breathing patterns elicited active sensorimotor connections with other ROIs. Uniquely, lateral region of sensorimotor area followed the homunculus that represents the face, including both the nose and mouth. Thus, the stimuli from both the nose and mouth induced sensorimotor network, like the previous study [19]. In addition to this, the connection of right and left of sensorimotor area appeared as the most significant connection (Table 1 and Table 3), concordant with previous results showing a significant correlation between the left and right sensorimotor area during rest [19,33].

### 4.3. Central Opercular Cortex

We found central opercular cortex and sensorimotor connection (Table 1). This region is the most significant seed that had multivariate connectivity in Mouth > Nose contrast (Table 2). The relation between mouth and central opercular cortex is known in previous studies [34,35]. Central opercular cortex were associated with oral movement and sensation [34]. It also contributes to the oral and tongue movement [35]. Opercular cortex has been reported to be significant for pharynx sensation and sensorimotor area had relation with breathing [32,34]. Therefore, we considered the connections of central opercular cortex and sensorimotor area as significant features of mouth breathing.

### 4.4. Lateral Parietal Cortex in DMN

The connection with DMN region appeared in the Mouth < Nose contrast, which was not included in the Mouth > Nose contrast. Because the majority of people breathe through their nose, nasal breathing could represent normal resting state FC. Significant functional connection with parietal operculum cortex and lateral parietal cortex in the DMN appeared in the Mouth < Nose contrast (Table 3). Furthermore, brainstem region only appeared in the Mouth < Nose contrast. Brainstem has the critical role of cardiac and respiratory function. Thus, nasal breathing might be a more reasonable condition for performing resting state.

### 4.5. Limitations

The recruited subjects in this study represent healthy young adults in their twenties. Considering that normal breathing is a basic life process, differences brought on by age and their impacts need to be investigated before forming generalizations. In addition, the number of subjects is lower than prior studies that assessed resting state fMRI readouts in healthy adults [36]. Therefore, we propose further studies with larger sample sizes and participants’ representative of a range of ages.

Since the variation in respiratory function and muscle activity according to mouth breathing is well known, mouth breathing-induced muscle activity and GLM analysis as task effects should be considered in further studies. With respect to the differences in respiratory volume and accessory muscle activation between nose and mouth breathing, additional physiological data are needed to understand the corresponding FC differences. This can be evaluated using correlation analysis between physiological data and FC.

Finally, our finding does not indicate resting state FC problems of mouth breathing syndrome. Future studies that define the interaction between cognitive decline and mouth breathing syndrome are required.

## 5. Conclusions

We hypothesized that mouth breathing has different effects on resting state FC than those caused by nasal breathing. Nasal breathing functional connections in the bilateral sensorimotor network and the DMN were similar to previous resting state fMRI results [19]. Most pairs were located around the sensorimotor area centered on the sensorimotor superior. On the other hand, mouth breathing has complex and diverse connection than nasal breathing. In addition, the ROI pairs were spread over the whole brain, and more strong connections were observed on the left than on the right. These difference connections in healthy subjects, even if the brain is resting, suggesting that this mouth breathing condition may have potential and various impacts on brain function. Increased FC or widespread changes of FC have been also observed at resting sate network connectivity of patients [37]. Therefore, the increased connectivity of the ROI pairs in mouth breathing shown in this study will be the underlying evidence defining the interaction between cognitive decline and mouth breathing syndrome. Therefore, authors could suggest that the conventional nasal breathing is more appropriate when the brain needs to rest, compared to mouth breathing according to the present study results.

## Figures and Tables

**Figure 1 brainsci-10-00704-f001:**
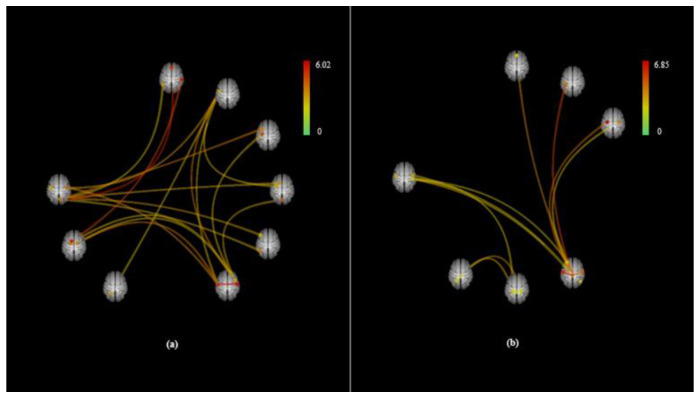
Region of interest (ROI)-to-ROI connectome ring maps of all selected ROI seeds in (**a**) Mouth > Nose and (**b**) Mouth < Nose contrasts. The color links are obtained at a ROI-to-ROI connections height threshold (FDR) of *P* < 0.05, seed ROI height threshold (uncorrected) of *P* < 0.05, with one-sided positive seed level correction and permutation tests. The color bar indicates the statistical T value. Abbreviations: ROI, region of interest; FDR, false discovery rate.

**Figure 2 brainsci-10-00704-f002:**
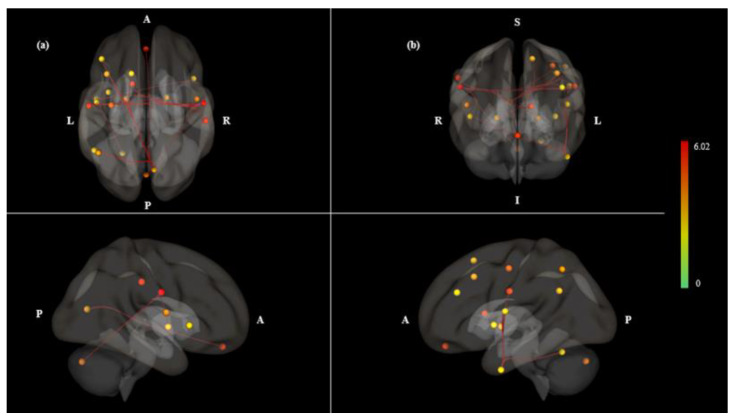
ROI-to-ROI 3D rendering maps of all selected ROI seeds in Mouth > Nose contrast, which was the same result as Figure 1a. 3D maps presented (**a**) dorsal, (**b**) anterior, (**c**) right and (**d**) left. The maps are obtained at a ROI-to-ROI connections height threshold (FDR) of *P* < 0.05, seed ROI height threshold (uncorrected) of *P* < 0.05, with one-sided positive seed level correction and permutation tests. The color bar indicates the statistical *T* value. Abbreviations: ROI, region of interest; FDR, false discovery rate.

**Figure 3 brainsci-10-00704-f003:**
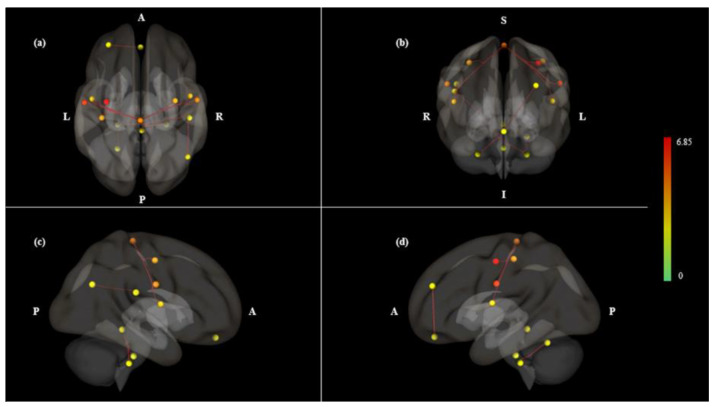
ROI-to-ROI 3D rendering maps of all selected ROI seeds in Mouth < Nose contrast, which was the same result as Figure 1b. 3D maps presented (**a**) dorsal, (**b**) anterior, (**c**) right and (**d**) left views. The maps are obtained at a ROI-to-ROI connections height threshold (FDR) of *P* < 0.05, seed ROI height threshold (uncorrected) of *P* < 0.05, with one-sided positive seed level correction and permutation tests. The color bar indicates the statistical *T* value. Abbreviations: ROI, region of interest; FDR, false discovery rate.

**Table 1 brainsci-10-00704-t001:** Mouth > Nose contrast ROI connections with threshold ROI to ROI connections by intensity at a 0.05 one-sided (positive) FDR-*P* value and threshold seed ROIs at a 0.05 uncorrected *P* value.

Contrast	Pair Connection	Statistics (*T*)	*P* Value	
			Uncorrected	FDR
Mouth > Nose	net.SM.Lateral (L)-net.SM.Lateral (R)	6.02	0.0000	0.0006
	net.SM.Lateral (R)-networks.SM.Lateral (L)	6.02	0.0000	0.0006
	SCC (R)-MedFC	5.49	0.0000	0.0018
	aSMG (R)-Caudate (L)	5.02	0.0000	0.0054
	aITG (L)-net.Salience.AInsula (R)	4.36	0.0002	0.0248
	net.SM.Lateral (L)-Pallidum (R)	4.34	0.0002	0.0113
	SCC (R)-MidFG (L)	4.3	0.0002	0.0142
	net.SM.Lateral (L)-CO (R)	4.22	0.0002	0.0113
	CO (R)-net.SM.Lateral (L)	4.22	0.0002	0.0339
	aITG (L)-CO (L)	3.87	0.0005	0.0389
	SCC (R)-net.FrontoParietal.LPFC (L)	3.85	0.0005	0.0269
	net.SM.Lateral (R)-net.Cerebellar.Posterior	3.78	0.0006	0.0425
	SCC (R)-SFG (L)	3.65	0.0008	0.0302
	SCC (R)-net.FrontoParietal.PPC (L)	3.58	0.0009	0.0302
	aITG (L)-IC (L)	3.56	0.001	0.0437
	net.SM.Lateral (L)-aITG (L)	3.52	0.0011	0.0437
	aITG (L)-net.SM.Lateral (L)	3.52	0.0011	0.0437
	net.SM.Lateral (R)-Pallidum (L)	3.51	0.0011	0.0425
	net.SM.Lateral (R)-PreCG (L)	3.46	0.0012	0.0425
	net.SM.Lateral (R)-Pallidum (R)	3.44	0.0013	0.0425
	aITG (L)-Cereb6 (L)	3.41	0.0014	0.0451
	SCC (R)-AG (L)	3.3	0.0018	0.0485

**Abbreviations**: ROI, region of interest; FDR, false discovery rate; R, right; L, left; net, networks; SM, sensorimotor; SCC, supracalcarine cortex; MedFC, frontal medial cortex; aSMG, supramarginal gyrus (anterior division); AInsula, anterior insula; MidFG, middle frontal gyrus; CO, central opercular cortex; aITG, inferior temporal gyrus (anterior division); LPFC, lateral prefrontal cortex; SFG, superior frontal gyrus; PPC, posterior parietal cortex; IC, insular cortex; PreCG, precentral gyrus; Cereb 6, cerebellum 6; AG, angular gyrus.

**Table 2 brainsci-10-00704-t002:** Mouth > Nose contrast connected regions included ROI that were more significantly connected at a 0.05 one-sided (positive) uncorrected-*P* value.

Area	Statistics	*P* Value
		Uncorrected
CO (R)	F(5)(16) = 6.73	0.0015
	Intensity = 4.22	0.0389
	Size = 1	0.0448
SM.Lateral (L)	F(5)(16) = 6.68	0.0015
	Intensity = 18.11	0.0037
	Size = 4	0.0055
SM.Lateral (R)	F(5)(16) = 6.05	0.0025
	Intensity = 20.21	0.0029
	Size = 5	0.0037
aITG (L)	F(5)(16) = 5.88	0.0029
	Intensity = 18.72	0.0035
	Size = 5	0.0037
SCC (R)	F(5)(17) = 4.83	0.007
	Intensity = 24.18	0.0021
	Size = 6	0.0026
aSMG (R)	F(5)(17) = 3.28	0.0315
	Intensity = 5.02	0.0194
	Size = 1	0.0448

**Abbreviations**: R, right; L, left; net, networks; CO, central opercular cortex; SM, sensorimotor; aITG, inferior temporal gyrus (anterior division); SCC, supracalcarine cortex; aSMG, supramarginal gyrus (anterior division).

**Table 3 brainsci-10-00704-t003:** Mouth < Nose contrast ROI connections with threshold ROI-to-ROI connections by intensity at a 0.05 one-sided (positive) FDR-*P* value and threshold seed ROIs at a 0.05 uncorrected *P* value.

Contrast	Pair Connection	Statistics (*T*)	*P* Value	
			Uncorrected	FDR
Mouth < Nose	net.SM.Lateral (L)-net.SM.Superior	6.85	0.0000	0.0001
	net.SM.Superior-net.SM.Lateral (L)	6.85	0.0000	0.0001
	net.SM.Superior–PostCG (L)	5.74	0.0000	0.0005
	net.SM.Superior–PreCG (L)	5.13	0.0000	0.0012
	net.SM.Superior-net.SM.Lateral (R)	5.06	0.0000	0.0012
	net.SM.Lateral (R)-net.SM.Superior	5.06	0.0000	0.0049
	net.Salience.RPFC (L)-MedFC	4.69	0.0001	0.0114
	Cereb10 (R)-Brain-Stem	4.67	0.0001	0.0121
	Cereb10 (R)-Vermis3	4.28	0.0002	0.0149
	Cereb10 (L)-Cereb6 (L)	4.13	0.0003	0.0421
	net.SM.Superior–CO (L)	3.97	0.0004	0.0123
	PO (R)-net.DefaultMode.LP (R)	3.93	0.0004	0.0465
	PO (R)-Cereb10 (L)	3.79	0.0006	0.0465
	Cereb10 (L)-PO (R)	3.79	0.0006	0.0465
	net.SM.Superior–CO (R)	3.77	0.0006	0.0163
	net.SM.Superior–PreCG (R)	3.7	0.0007	0.0166

**Abbreviations**: ROI, region of interest; FDR, false discovery rate; R, right; L, left; net, networks; SM, sensorimotor; PostCG, postcentral gyrus; PreCG, precentral gyrus; RPFC, rostral prefrontal cortex; MedFC, frontal medial cortex; Cereb10, cerebellum 10; Cereb 6, cerebellum 6; CO, central opercular cortex; PO, parietal operculum cortex; LP, lateral parietal cortex.

**Table 4 brainsci-10-00704-t004:** Mouth < Nose contrast connected regions included ROI that were more significantly connected at a 0.05 one-sided (positive) uncorrected-*P* value.

Area	Statistics	*P* Value
		Uncorrected
SM.Lateral (L)	F(5)(16) = 6.68	0.0015
	Intensity = 6.85	0.0168
	Size = 1	0.0458
SM.Lateral (R)	F(5)(16) = 6.05	0.0025
	Intensity = 5.06	0.0191
	Size = 1	0.0458
SM.Superior	F(5)(16) = 5.63	0.0035
	Intensity = 34.21	0.0009
	Size = 7	0.0019
PO (R)	F(5)(16) = 4.25	0.012
	Intensity = 7.72	0.0165
	Size = 2	0.0167
Cereb10 (L)	F(5)(16) = 3.7	0.0204
	Intensity = 7.93	0.0155
	Size = 2	0.0167
Cereb10 (R)	F(5)(16) = 2.98	0.0435
	Intensity = 8.95	0.0105
	Size = 2	0.0167
Salience.RPFC	F(5)(16) = 3.53	0.0242
	Intensity = 4.69	0.0232
	Size = 1	0.0458

**Abbreviations**: ROI, region of interest; R, right; L, left; net, networks; SM, sensorimotor; PO, parietal operculum cortex; Cereb10, cerebellum 10; RPFC, rostral prefrontal cortex.

**Table 5 brainsci-10-00704-t005:** Lateralization index on Mouth > Nose contrast in each hemisphere.

Left Pair Connection	Statistics (*T*)	Right Pair Connection	Statistics (*T*)	L + R	L-R	LI
aITG (L)-CO (L)	3.87	aITG (R)-CO (R)	1.76	5.63	2.11	0.375 *
aITG (L)-IC (L)	3.56	aITG (R)-IC (R)	0.92	4.48	2.64	0.589 *
net.SM.Lateral (L)-aITG (L)	3.52	net.SM.Lateral (R)–aITG (R)	1.39	4.91	2.13	0.434 *
aITG (L)-Cereb6 (L)	3.41	aITG (R)-Cereb6 (R)	0.75	4.16	2.66	0.639 *
net.SM.Lateral (L)-Pallidum (L)	3.31	net.SM.Lateral (R)-Pallidum (R)	3.44	6.75	−0.13	−0.019

Significant left lateralization (LI > 0.2) was expressed as *. **Abbreviations**: R, right; L, left; aITG, inferior temporal gyrus (anterior division); CO, central opercular cortex; IC, insular cortex; net, networks; SM, sensorimotor; Cereb 6, cerebellum 6.

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
