# Peer review of "Investigation of Functional Connectivity Differences between Voluntary Respirations via Mouth and Nose Using Resting State fMRI"

_brainsci, 2020, doi:10.3390/brainsci10100704_

Round 1

Reviewer 1 Report

This study investigates the differences in resting-state functional connectivity (FC) collected during two types of breathing (nose versus mouth). Resting-state fMRI was collected in 22 young healthy individuals. Overall, authors reported stronger FC during the mouth breathing than during the nose breathing condition.

Despite a first round of revision, I think more major clarifications are needed – especially in the method and results. The large majority of my comments is related to unclear sentences that may benefit from the involvement of an English-Native Speaker. I listed all my questions and comments below.

Abstract:

Authors mention 23 participants – it is 22 in the method. Please correct as appropriate.

“the region of interest analyses toolbox” suggests that it is the name of the toolbox. While I understand that the authors rephrased based on a reviewer’s comment, there are other ways to write this sentence to make it clear and accurate without naming a toolbox.

I think the last sentence is a clear over-statement. There is no evidence of “anomalous connections” – just differences within a healthy group of participants. No associations with cognitive impairments have been described in this study as well. Please delete this sentence and/or rephrase your interpretations.

“respiratory FC” is an incorrect statement, please delete “respiratory”. Also I don’t think that the results suggest that FC is “more affected” by mouth breathing. Again, it is just differences reported between two conditions.

Methods:

Section 2.1: line 87: ”All participants underwent two imaging sequences in scanner.” Should it rather be three sequences? One structural and 2 Resting-state.

Section 2.2: Lines 134+: This paragraph needs more detail – I am suggesting to have its own section. Please describe how many ROIs / How many RSNs were investigated. Why describe the DMN and not the other networks? It is unclear whether authors investigated the FC between regions within each RSN or FC between RSN, etc. Were the analyses between ROI related to the RSNs or totally independent from the RSN?

Section 2.3: Title is incorrect. “Laterality of FC” might be more appropriate.

Section 2.4: Where are the results from the seed-based analyses reported? It seems incorrect. If authors really did such analyses, why are they not listed with the ROIs analyses and what seed was used (and provide a justification in this case)? An analysis that focused on a ROI vs multiple other ROIs is not a seed based analysis. Please rephrase / reorganize your statistical analyses sections in order to help the reader to understand what exact analyses were done and in which order. Same comment throughout the manuscript when the word “seed” is used. Also, how can the authors apply some sort of “seed-level correction” when there is no whole brain analysis done? Authors should correct based on the number of comparisons done.

Line 158: “The 3D rendering ….” Should be moved to a relevant section (like in the legend of the figure only) and should be rephrased: “for understanding the FC” is awkward. Furthermore, authors should talk about dorsal and ventral view rather than superior and inferior.

Figure 1 is very unclear. Some lines seem to go to no brain. Also, why are they multiple regions on the brain pictures? Which line to which region? I also suggest to change the colormap/scale. All the colors seem to be orange. Is the color of the ROIs relate to the color of the lines?

I don’t quite understand how Figures 1, 2 and 3 are different? Do they show different levels of analyses (ROI and RSNs)? If so it should be clear in the legend and the text.

I honestly do not understand Tables 2 and 4. What are the statistics done on? What are “intensity” and “size” referring to? Also, I suggest to report the actual mean (sd) FC between the mouth vs nose breathing conditions (although it might be more appropriate for Table 1).

Author Response

Authors’ Responses to Reviewers’ Comments:

Thank you very much for your valuable comments and suggestions. We carefully revised the manuscript following the reviewers’ recommendations accordingly. Point-by-point responses are as follows:

Reviewer's Comments to the Author:

-----------------------------------------------------------------------------------------------------

Reviewer #1:

This study investigates the differences in resting-state functional connectivity (FC) collected during two types of breathing (nose versus mouth). Resting-state fMRI was collected in 22 young healthy individuals. Overall, authors reported stronger FC during the mouth breathing than during the nose breathing condition.

Despite a first round of revision, I think more major clarifications are needed – especially in the method and results. The large majority of my comments is related to unclear sentences that may benefit from the involvement of an English-Native Speaker. I listed all my questions and comments below.

Abstract:

Authors mention 23 participants – it is 22 in the method. Please correct as appropriate.

R1_1: Twenty-two healthy subjects participated in the study but one subject was excluded for the analysis due to the large global signals (GS) variance. Therefore, data of 21 subjects were analyzed in this study, so we corrected it appropriately. Thank you for your comment.

“the region of interest analyses toolbox” suggests that it is the name of the toolbox. While I understand that the authors rephrased based on a reviewer’s comment, there are other ways to write this sentence to make it clear and accurate without naming a toolbox.

R1_2: Thank you for your recommendation. We have modified the phrase to read: “The region of interest (ROI) analysis of functional connectivity in fMRI was conducted using a Matlab-based imaging software”

I think the last sentence is a clear over-statement. There is no evidence of “anomalous connections” – just differences within a healthy group of participants. No associations with cognitive impairments have been described in this study as well. Please delete this sentence and/or rephrase your interpretations.

R1_3: We replaced “anomalous connections” with “differences of connection” as the reviewer’s suggestion. The cognitive impairments part also changed appropriately.

“respiratory FC” is an incorrect statement, please delete “respiratory”. Also I don’t think that the results suggest that FC is “more affected” by mouth breathing. Again, it is just differences reported between two conditions.

R1_4: We corrected “respiratory FC” with “FC difference between two breathing conditions” and deleted the expression of “more affected” as the reviewer’s recommendation.

Methods:

Section 2.1: line 87: ”All participants underwent two imaging sequences in scanner.” Should it rather be three sequences? One structural and 2 Resting-state.

R1_5: The reviewer was right. The used sequences were two, but three scans (one structural and 2 resting-state scans) were conducted for each subject. We have clarified the expression to address the reviewer’s concern.

Section 2.2: Lines 134+: This paragraph needs more detail – I am suggesting to have its own section. Please describe how many ROIs / How many RSNs were investigated. Why describe the DMN and not the other networks? It is unclear whether authors investigated the FC between regions within each RSN or FC between RSN, etc. Were the analyses between ROI related to the RSNs or totally independent from the RSN?

R1_6: FC analysis was conducted with a total of 164 ROIs. Among them, 132 areas of FSL Harvard-Oxford atlas and Automated Anatomical Labelling (AAL) atlas were included, and 32 atlas related to network were included (i.e., 32 ROIs across 8 networks). The total of 8 networks used were DMN, sensorimotor network, visual networks, salience networks, dorsal attention networks, frontoparietal networks, language networks, cerebellar networks. RSN and ROIs were each independently analyzed, and functional connectivity pairs were obtained with 164 ROIs in which all of the RSN ROIs and atlas ROIs were combined. The reason why only DMN was described in the method was representatively because it was judged that DMN was most commonly known in RSN and that the lateral parietal region of DMN had a significant relationship with the results of this study.

Section 2.3: Title is incorrect. “Laterality of FC” might be more appropriate.

R1_7: It looks much better. Thank you for your recommendation.

Section 2.4: Where are the results from the seed-based analyses reported? It seems incorrect. If authors really did such analyses, why are they not listed with the ROIs analyses and what seed was used (and provide a justification in this case)? An analysis that focused on a ROI vs multiple other ROIs is not a seed based analysis. Please rephrase / reorganize your statistical analyses sections in order to help the reader to understand what exact analyses were done and in which order. Same comment throughout the manuscript when the word “seed” is used. Also, how can the authors apply some sort of “seed-level correction” when there is no whole brain analysis done? Authors should correct based on the number of comparisons done.

R1_8: First, seed-based analysis was performed in the first level analysis for each subject to obtain the result of connectivity between individual seed to voxel or ROI. The seeds used at this time were equally applied to those within 164 ROIs. Second, the second level analysis between groups was performed to analyze the contrast between conditions. In this study, only ROI to ROI results were analyzed in the second level analysis, so only ROI to ROI results were presented in the final results.

Line 158: “The 3D rendering ….” Should be moved to a relevant section (like in the legend of the figure only) and should be rephrased: “for understanding the FC” is awkward. Furthermore, authors should talk about dorsal and ventral view rather than superior and inferior.

R1_9: Thank you for your valuable comments. We have corrected the words and moved the sentence to the results section.

Figure 1 is very unclear. Some lines seem to go to no brain. Also, why are they multiple regions on the brain pictures? Which line to which region? I also suggest to change the colormap/scale. All the colors seem to be orange. Is the color of the ROIs relate to the color of the lines?

R1_10: Figure 1 was set to display 12 reference brains provided by CONN, with all the ROI rings overlaid on the reference. Because several ROI regions were arranged along the ring, they could appear together in the reference brains. However, we modified the colors of the ROIs and the color map to make them more distinct.

I don’t quite understand how Figures 1, 2 and 3 are different? Do they show different levels of analyses (ROI and RSNs)? If so it should be clear in the legend and the text.

R1_11: Thank you for your valuable comments. Figure 1 shows the connection ring method to intuitively view the pair between the ROIs. Figures 2 and 3 show the ROI connectivity in 3D space using a 3D display method to allow spatial understanding of the location and connectivity of the ROIs. And, we have corrected the figure legends like ‘Figure 2. ROI-to-ROI 3D rendering maps of all selected ROI seeds in Mouth > Nose contrasts shown in Figure 1(a) presented in (a) dorsal, (b) anterior, (c) right and (d) left view.

I honestly do not understand Tables 2 and 4. What are the statistics done on? What are “intensity” and “size” referring to? Also, I suggest to report the actual mean (sd) FC between the mouth vs nose breathing conditions (although it might be more appropriate for Table 1).

R1_12: In Tables 2 and 4, the statistical results were thresholded using a seed level thresholds (F-test on the multivariate connectivity strength for each seed ROI). The size measurement represents the number of above-threshold connections between the seed ROI and all other ROIs, and the intensity represents the overall strength as the sum of the absolute T-values for these above-threshold connections (Mckenna et al., 2016).

Reviewer 2 Report

This study assessing the novel question of brain connectivity during mouth vs. nose breathing at rest. While the manuscript is quite clear overall, my main concern is regarding the inadequate level of detail provided regarding the methods employed, as this makes the results difficult to interpret for readers who are not familiar with the CONN toolbox. Other than this I only have minor comments, and found the manuscript and results highly interesting.

Minor comments:

  • Abstract (line 27): The discussion of the findings does not match the reported results – this study does not assess learning rates and cognitive decline, and thus this should not be included in the abstract as it is speculation.
  • Introduction (line 64): I would advise against making statements comparing the utility of resting-state vs. task-based experiments and analyses. This is a matter of opinion and is not needed for this paper – the use of resting state is perfectly acceptable without this. I would suggest removing this paragraph.
  • Methods (line 139): Why was the laterality analysis not conducted on the Mouth < Nose contrast as well?
  • Methods (line 146): There is currently not enough detail provided in the methods section to understand the difference between the analyses conducted. How are the ROI-to-ROI and seed-to-voxel analyses conducted? How are these different? Without this information it is very difficult to interpret the results for any reader that is not familiar with the CONN toolbox, e.g. It is not currently clear what the results presented in Tables 2 and 4 are with the current level of detail provided.
  • Results (line 223): If the laterality results are going to be mentioned in the abstract and discussion as one of the main results of the study, then the Table of results should be included in the main manuscript rather than the supplementary material.
  • Discussion (line 250): The first sentence of the discussion suggests that a brain function analysis was performed, rather than only a functional connectivity analysis. Please correct this.

Author Response

Authors’ Responses to Reviewers’ Comments:

Thank you very much for your valuable comments and suggestions. We carefully revised the manuscript following the reviewers’ recommendations accordingly. Point-by-point responses are as follows:

Reviewer's Comments to the Author:

-----------------------------------------------------------------------------------------------------

Reviewer #2:

This study assessing the novel question of brain connectivity during mouth vs. nose breathing at rest. While the manuscript is quite clear overall, my main concern is regarding the inadequate level of detail provided regarding the methods employed, as this makes the results difficult to interpret for readers who are not familiar with the CONN toolbox. Other than this I only have minor comments, and found the manuscript and results highly interesting.

Minor comments:

Abstract (line 27): The discussion of the findings does not match the reported results – this study does not assess learning rates and cognitive decline, and thus this should not be included in the abstract as it is speculation.

R2_1: We rephrased the sentence as the reviewer commented. Increased functional connectivity or widespread changes of functional connectivity have been also observed at resting sate network connectivity of patients (Chenji et al., 2016). Therefore, the increased connectivity of the ROI pairs in mouth breathing shown in this study will be the underlying evidence defining the interaction between cognitive decline and mouth breathing syndrome. But neuroscientific as well as clinical evidences for mouth breathing should be performed in future studies.

Introduction (line 64): I would advise against making statements comparing the utility of resting-state vs. task-based experiments and analyses. This is a matter of opinion and is not needed for this paper – the use of resting state is perfectly acceptable without this. I would suggest removing this paragraph.

R2_2: We removed the content based on the comments of the reviewer and corrected the paragraph accordingly.

Methods (line 139): Why was the laterality analysis not conducted on the Mouth < Nose contrast as well?

R2_3: In Mouth> Nose contrast, the connection in the left intrahemisphere was much more prominent than that in the right, so the evaluation for the left laterality was performed. In contrast, in Mouth <Nose contrast, since one connection in the left (Cereb10 (L)-Cereb6 (L )) and one connectivity in the right (PO (R)-net.DefaultMode, LP(R)) was found, it was concluded that the evaluation of laterality was not significant.

Methods (line 146): There is currently not enough detail provided in the methods section to understand the difference between the analyses conducted. How are the ROI-to-ROI and seed-to-voxel analyses conducted? How are these different? Without this information it is very difficult to interpret the results for any reader that is not familiar with the CONN toolbox, e.g. It is not currently clear what the results presented in Tables 2 and 4 are with the current level of detail provided.

R2_4: We refined the method in detail as suggested by the reviewer. Please also refer to the above author responses (R1_8 and R1_12).

Results (line 223): If the laterality results are going to be mentioned in the abstract and discussion as one of the main results of the study, then the Table of results should be included in the main manuscript rather than the supplementary material.

R2_5: We included the laterality results in the main text as suggested by the reviewer.

Discussion (line 250): The first sentence of the discussion suggests that a brain function analysis was performed, rather than only a functional connectivity analysis. Please correct this.

R2_6: We have modified the sentence. Thank you for the reviewer’s recommendation.

Reviewer 3 Report

The topic is significant as there is much interest in the benefits of nasal vs. mouth breathing. The interpretation of the data is difficult due to lack of explanation of the connectivity measures, and the omission of negative findings, specifically which VOI were studied and found to not have significant effects. Most of the discussion is helpful, and reviewing the regional differences in particular is helpful. The data collection and analysis methods are appropriate, with one concern mentioned below. I appreciate the authors have already responded to previous reviewers’ comments, but I believe more changes are needed to help readers understand the meaning of these findings.

One issue is the subject who had a global signal that was an outlier (and excluded from Figure S1); how do the authors justify keeping the participant in the main analysis? How might this large global signal change affect the results? Was the large signal change in both nasal and mouth breathing fMRI series, or just one? If the former, the participant could add noise. If the latter, the participant could bias the findings.

The general question I have is how to relate the findings to brain function. How are we to interpret the greater connectivities in mouth vs nasal from a physiological perspective? What brain areas are these in, and what do they relate to? What does symmetrical connectivity mean? In the abstract, the authors conclude “These 27 anomalous connections might correspond to lower learning ability and work efficiency,” but this correspondence is unclear. How would recruiting stronger connections between some brain regions (with mouth breathing) lead to poorer brain function? Surely the brain is wired to have connections?

Since the DMN is defined by existing rsfMRI studies, the DMN presumably reflects nasal breathing since the majority of people breathe through their nose while lying in the MRI scanner. This should be noted in the discussion.

Introduction: it’s not clear why functional connectivity is important – what does it mean if there are different connectivities between nasal and mouth breathing? Is that good, bad, different? Since this is a journal with a general neuroscience audience, as opposed to just people familiar with neuroimaging, some explanation related to neural function would be helpful.

L43, mouth breathing syndrome is not the same as mouth breathing

L54 Of citations 10-14, only citation 12 relates to the brain

L109 Give a brief overview of what is being analyzed, so the reader understands the point o

L134 The atlases should be cited, and ideally the VOI should be listed. Perhaps a table in the Supplemental file? This is important since the findings of regions that did not show significant connectivity are equally.

L140 What does “right lateralization” mean in terms of connectivity? One region with more right connections? More inta-hemisphere connections? Need more explanation.

L167 Looking at Figures 2 and 3, it is not obvious there is any major difference between the conditions. What should the reader be looking for in these figures?

L169 What VOI were tested? What seeds were used? The negative findings are also important.

L180 Need labels to be able to interpret what regions are connected.

L187, 193 – What were selected ROI?

L199, 209 What ROI were tested and had non-significant effects?

L260 How does language processing relate to breathing through the nose or mouth?

L280 “the stimuli from both the nose and mouth might affect the sensorimotor network” – this is known, it is what defines the sensorimotor network.

Author Response

Authors’ Responses to Reviewers’ Comments:

Thank you very much for your valuable comments and suggestions. We carefully revised the manuscript following the reviewers’ recommendations accordingly. Point-by-point responses are as follows:

Reviewer's Comments to the Author:

-----------------------------------------------------------------------------------------------------

Reviewer #3:

The topic is significant as there is much interest in the benefits of nasal vs. mouth breathing. The interpretation of the data is difficult due to lack of explanation of the connectivity measures, and the omission of negative findings, specifically which VOI were studied and found to not have significant effects. Most of the discussion is helpful, and reviewing the regional differences in particular is helpful. The data collection and analysis methods are appropriate, with one concern mentioned below. I appreciate the authors have already responded to previous reviewers’ comments, but I believe more changes are needed to help readers understand the meaning of these findings.

One issue is the subject who had a global signal that was an outlier (and excluded from Figure S1); how do the authors justify keeping the participant in the main analysis? How might this large global signal change affect the results? Was the large signal change in both nasal and mouth breathing fMRI series, or just one? If the former, the participant could add noise. If the latter, the participant could bias the findings.

R3_1: We re-analyzed the data without the outlier. We have corrected the manuscript accordingly, including Table and text. Total 21 participants (except for one subject due to GS outlier) results were presented at the manuscript (Figures 1,2 and 3; Tables 1,2,3,4 and 5). 

The general question I have is how to relate the findings to brain function. How are we to interpret the greater connectivities in mouth vs nasal from a physiological perspective? What brain areas are these in, and what do they relate to? What does symmetrical connectivity mean? In the abstract, the authors conclude “These 27 anomalous connections might correspond to lower learning ability and work efficiency,” but this correspondence is unclear. How would recruiting stronger connections between some brain regions (with mouth breathing) lead to poorer brain function? Surely the brain is wired to have connections?

R3_2: In this study, we found that mouth breathing showed more and stronger functional connectivity than nose breathing. A strong connection may not cause weak brain function, but it may not be a phenomenon seen in normal people. This increase in the resting state functional connectivity is also a phenomenon that occurs in the abnormal FC. It has been reported that more widespread changes in functional connectivity are observed in patients with disease (Chenji et al., 2016). As such, it is necessary to study what effect the increased FC can have in mouth breathing, and also to investigate the association with cognitive impairment in mouth breathing syndrome.

In Nose > Mouth contrast, symmetrical connectivity was remarkably observed. Symmetrical connectivity means that one seed (sensorimotor superior) and its connected ROIs were observed symmetrically in both hemispheres. The sensorimotor superior was the seed most connected to other ROIs at Nose>Mouth, it was observed that ROIs of both hemispheres were correlated at the same time. In Figure 1, it was seen that the preCG(L), preCG(R), sensorimotor lateral(L), sensorimotor lateral(R), CO(L), and CO(R) regions were connected to the sensorimotor superior in both hemispheres. In the resting state networks of normal subjects, it was observed that functional links were symmetrically formed in both hemispheres (Heuvel et al.. 2010), and it is considered that similar patterns were observed in Nose >Mouth as described above.

Finally, functional connectivity reflects direct interaction between two regions even though there may exist between anatomically unconnected regions (Batista-García-Ramó et al., 2018). That is, FC can appear even if the fiber is not connected.  

Figure. The coronal view FCs of the sensorimotor superior in Nose > Mouth contrast.

Since the DMN is defined by existing rsfMRI studies, the DMN presumably reflects nasal breathing since the majority of people breathe through their nose while lying in the MRI scanner. This should be noted in the discussion.

R3_3: Authors agreed with the reviewer’s comment. We have mentioned it in the discussion session.  

Introduction: it’s not clear why functional connectivity is important – what does it mean if there are different connectivities between nasal and mouth breathing? Is that good, bad, different? Since this is a journal with a general neuroscience audience, as opposed to just people familiar with neuroimaging, some explanation related to neural function would be helpful.

R3_4: A suitable resting of the brain is an important process for recovery and activation of the brain, and breathing is an important factor that can influence the resting state. The results of this study proved that the functional connectivity of mouth breathing in the resting state was different from that of nasal breathing. ROI Connections in mouth breathing suggested that brain activity different from that of conventional nasal breathing could be induced, which means that brain activity in the resting state of the brain may be different.

As the resting state is an important process responsible for energy consumption of 70-80% of the brain, it is thought that the brain activity changed in the resting state can also affect the energy metabolism of the brain. Therefore, further study should be followed to determine if metabolic changes occur during mouth breathing.

L43, mouth breathing syndrome is not the same as mouth breathing

R3_5: We used mouth breathing syndrome to describe the effects of various mouth breathing, including otolaryngological symptoms. Therefore, we corrected the expression appropriately.

L54 Of citations 10-14, only citation 12 relates to the brain

R3_6: We included the references to describe the current studies, mainly focusing on the structural effects of mouth breathing, i.e., on the otolaryngology. We removed some of them and added studies of brain function by mouth breathing.

L109 Give a brief overview of what is being analyzed, so the reader understands the point o

R3_7: We have further described the analysis method as the reviewer suggested. And please also refer to the above author responses (R1_8 and R1_12).

L134 The atlases should be cited, and ideally the VOI should be listed. Perhaps a table in the Supplemental file? This is important since the findings of regions that did not show significant connectivity are equally.

R3_8: We listed the atlas in the Supplemental file as the reviewer suggested.

L140 What does “right lateralization” mean in terms of connectivity? One region with more right connections? More inta-hemisphere connections? Need more explanation.

R3_9: Lateralization means having a higher hemisphere pairwise connection strength than the other hemisphere. We compared the connectivity strengths between the left and right hemisphere of aITG (L) – Cereb6 (L) and aITG (R) – Cereb6 (R), respectively. For example, the superior value of the left hemisphere was expressed by the left laterality. In this study, the left pairwise hemisphere connection was observed more prominently than the right hemisphere in Mouth> Nose contrast, and then the statistical analysis for this laterality was performed to prove the left laterality. However, right lateralization was not observed at Mouth>Nose contrast.

L167 Looking at Figures 2 and 3, it is not obvious there is any major difference between the conditions. What should the reader be looking for in these figures?

R3_10: When comparing Figures 2 and 3, we can find that ROI to ROI connections in Figure 2 (Mouth > Nose) are more complex and diverse than those in Figure 3. The number of ROI pairs in Figure 2 was more than that in Figure 3. In addition, the ROI pairs in Figure 2 were spread over the whole brain, and more stronger connection were observed on the left than on the right, whereas in Figure 3, it was seen that the pairs were located around the sensorimotor area centered on the sensorimotor superior.

L169 What VOI were tested? What seeds were used? The negative findings are also important.

R3_11: FC analysis was conducted with a total of 164 ROIs. Among them, 132 areas of FSL Harvard-Oxford atlas and Automated Anatomical Labelling (AAL) atlas were included, and 32 atlas related to network were included (i.e., 32 ROIs across 8 networks). We listed the atlas in the Supplemental file as the reviewer suggested. Please also refer to the above author response (R1_6). Mouth > Nose contrast and Nose > Mouth contrast has an opposite relation. Therefore, the negative functional correlation of Mouth > nose contrast is Nose > Mouth contrast positive functional correlation.

L180 Need labels to be able to interpret what regions are connected.

R3_12: Figure 1 showed only the connection lines to easily express the difference in the number of connections in Mouth>Nose contrast and Nose>Mouth contrast. Since it was more difficult to observe when labels were presented for too many connections, the connected areas were separately specified in Tables 1 and 3. Please refer to Tables 1 and 3 for all the connected regions.

L187, 193 – What were selected ROI?

R3_13: Figures 2 and 3 were the results of analyzing 164 ROIs and the selected ROI meant the seeds provided in Tables 2 and 4. The areas in Tables 2 and 4 were the selected ROIs.

L199, 209 What ROI were tested and had non-significant effects?

R3_14: All 164 ROIs were tested and the significant results were provided in Tables 1 and 2. Statistically non-significant results were not displayed. Note that all the ROIs were listed at the supplementary file (Table S1)

L260 How does language processing relate to breathing through the nose or mouth?

R3_15: When speaking or vocalizing, mouth breathing is accompanied according to the phrase, and exhalation is performed through the oral cavity (Moulin-Frier et al., 2014). To maintain breathing, inhalation is performed through the oral cavity or nasal cavity. Furthermore, the speech processing has a lot of influence in the left hemisphere as it is involved in Broca and Wernicke (Knecht et al., 2000; Marc et al., 2016). It is thought that stimulation by oral breathing induces connectivity of the left hemisphere like speech. According our results, left inferior temporal gyrus connections which have significantly left lateralized connections were appeared in Mouth > Nose contrast (Table 5). Left inferior temporal gyrus known as important structure for language process had linked with left central opercular cortex and left insular cortex which also have a role in language processing.    

L280 “the stimuli from both the nose and mouth might affect the sensorimotor network” – this is known, it is what defines the sensorimotor network.

R3_16: We have corrected the sentence accordingly.

Round 2

Reviewer 1 Report

Authors appropriately responded to my concerns. I do not have further comments. 

Reviewer 3 Report

I think the changes address my comments to a reasonable degree, and with some final language editing this is suitable for publishing.

Thanks for the opportunity to review this.

This manuscript is a resubmission of an earlier submission. The following is a list of the peer review reports and author responses from that submission.

Round 1

Reviewer 1 Report

This paper by Jung et al is well written and on an important topic. 

My primary concern is that physiological recordings do not appear to have been collected.  If this information is available, it would be instructive to see the respiratory traces for all subjects, to confirm that there are no systematic differences between the two conditions (nasal, mouth) with respect to the rate and depth of breathing. If this information is not available, this would be a significant limitation of the study in my view.

Why? As the authors know I am sure, breathing itself influences cerebral blood flow (by altering the concentration of C02 in arterial blood), which in turn affects resting-state fMRI signals. Certain kinds of breathing patterns (such as "deep breaths" or "sighs"; see Power et al. 2019 NeuroImage for example) influence fMRI signals globally, but the effects tend to be strongest in many brain regions implicated in this investigation (including sensorimotor cortex). For this reason, without respiratory recordings, in my view, it makes interpretation of some findings in this manuscript difficult. 

If respiratory recordings for these participants were not collected, one way to address this issue may be to test for differences in global signal (GS) variance between nasal breathing and mouth breathing conditions. In other words, the authors could calculate the standard deviation of the average gray matter time-course for all subjects and compare nasal GS variance and mouth GS variance.

In summary, understanding the effects of respiration and different kinds of breathing patterns (and now breathing strategies; e.g., nasal vs. mouth) is an important topic and understanding how these factors might bias functional connectivity studies is important. I hope that the authors are able to obtain respiratory recordings from these participants and if not, are able to perform the suggest global signal analysis. Thank you for the opportunity to review this work. 

Reviewer 2 Report

This works examines functional connectivity differences between nasal and mouth breathing. I think the idea is interesting for the community. I have some further comments that I believe need to be addressed prior to decision for publication.

Abstract:

  1. The very first sentence in the abstract says, ‘The variations in respiratory function and muscle activity between mouth and nasal breathing methods have been well-studied, but the neural correlates of functional connectivity (FC) still remain unclear.’. I think the main limitation of the study is that they just focus on the latter part, although they immediately mention the importance of resp-function and muscle activity.

  1. CONN toolbox should be described in the abstract. Or authors can just mention ROIs analyses were performed, and omit use of acronym CONN.

Introduction:

  1. P2, l54: Increased oxygen load is an interesting finding during mouth inhalation, although if this would directly affect the amount of oxygen transferred to the lungs is not clear. I also wonder how it the exchange in CO2 during mouth versus nasal exhalation? Is there any literature on that?

Methods:

  1. P3, l111: ‘We used an established strategy for spatial and temporal preprocessing to define and eliminate confounds in the BOLD signal to prevent the effects of movement and physiological noise factors in the data [18].’ Could you explain briefly what is the approach used here?
  2. P3, l115: How many motion parameters were included in the regression?
  3. P4, l144:So how was the multiple comparison was performed? Appropriately corrected is not so clear.

Results:

  1. There is probably a huge respiratory variation effect during mouth breathing, as I do expect an increased respiratory volume, it may almost induce a deeper breathing than usual. I suggest including a potential deeper inspiration effect in discussion, and comparing if any literature exists on FC results with deep breathing (mouth or nasal).
  2. I also think mouth breathing is almost a 'task' case versus nasal breathing. I also suggest authors to include this as a task in a GLM analyses, and repeat their results after regressing 'task' effects, and then compare to nasal breathing. If you had RV data, that could be regressed out as well.
  3. Figure 2: Are there any regions with negative T values?

Discussion:

  1. P11, l230: ‘Although the Mouth > Nose contrast showed left lateralized asymmetry patterns.’ What is the rationale behind that? It is mentioned in the draft that, these regions are more related to respiratory areas. But would be great if authors could discuss that in more detailed. Especially lines 239-241 need references
  2. P12, 274-276: I guess this was a note from an earlier version?

Conclusions:

  1. P12, l291: Could you explain those prominent effects? Maybe point to the finding for me?